# Effects of Antioxidant Vitamins, Curry Consumption, and Heavy Metal Levels on Metabolic Syndrome with Comorbidities: A Korean Community-Based Cross-Sectional Study

**DOI:** 10.3390/antiox10050808

**Published:** 2021-05-19

**Authors:** Hai Nguyen Duc, Hojin Oh, Min-Sun Kim

**Affiliations:** Department of Pharmacy, College of Pharmacy and Research Institute of Life and Pharmaceutical Sciences, Sunchon National University, Sunchon 57922, Korea; haiyds@gmail.com (H.N.D.); navytoto@hanmail.net (H.O.)

**Keywords:** antioxidant vitamins, heavy metals, curry consumption, metabolic syndrome, comorbidities

## Abstract

The burden of metabolic syndrome (MetS) has increased worldwide, especially during the COVID-19 pandemic, and this phenomenon is related to environmental, dietary, and lifestyle risk factors. We aimed to determine the association between the levels of serum heavy metals, hs-CRP, vitamins, and curry intake and to predict risks of MetS based on marginal effects. A data set of 60,256 Koreans aged ≥ 15 years between 2009 and 2017 was used to obtain information on sociodemographic, lifestyle, family history characteristics, MetS, food intake survey, and serum heavy metals. Daily intake of vitamins was measured by a one-day 24 h recall, and curry consumption was calculated using a food frequency questionnaire. Serum heavy metal levels were quantified by graphite furnace atomic absorption spectrometry and using a mercury analyzer. We found that vitamin B1, B2, B3, C, and A intakes were significantly lower in subjects with than without MetS. In contrast, serum levels of Pb, Hg, Cd, vitamin A, E, and hs-CRP were significantly higher in subjects with MetS. The risk of MetS was significantly lower for high curry consumers than low curry consumers (adjusted odds ratio 0.85, 95%CI 0.74–0.98). The risks of MetS were reduced by 12% and 1%, when vitamin B1 and C intakes increased by one mg, respectively, but were increased by 14%, 3%, and 9%, when serum levels of Pb, Hg, and hs-CRP increased by one unit. These results show that the potential health benefits resulting from vitamin and curry intakes could protect the public against the dual burden of communicable and non-communicable diseases. Further studies are required to reduce risk factors associated with serum heavy metal levels and to determine whether interactions between vitamin and curry consumption influence the presence of MetS.

## 1. Introduction

Food is now understood to be a significant modifiable contributor to chronic disease, and empirical evidence indicates that dietary modifications may have positive or negative impacts on lifelong health [1]. Consumption of high-saturated-fat and high-energy diets, overuse of tobacco and alcohol, and sedentary lifestyles have contributed to increases in non-communicable diseases (NCDs), especially metabolic syndrome (MetS) [1,2]. MetS is a collection of metabolic disorders—that is, insulin resistance, dyslipidemia, central obesity, and hypertension—and a risk factor for the development of type 2 diabetes and cardiovascular diseases [3]. Of note, recent evidence suggests that MetS affects the progression and prognosis of COVID-19, and its severity has been correlated with poorer COVID-19 outcomes [4,5].

In addition to lifestyle and genetic factors, heavy metals are also risk factors of MetS [6,7,8,9]. Levels of heavy metals, especially lead (Pb), mercury (Hg), and cadmium (Cd), released into the environment by vehicles or factories or in contaminated seafood are increasing and accumulate in bones, kidney cortices, and lungs [10]. Heavy metals catalyze the release of reactive oxygen species (ROS), inflammatory mediators, and antithrombotic substances that damage vascular endothelial cells and exacerbate hypertension [7,11]. In particular, Pb and Cd disrupt blood clotting and increase the risk of CVD [7,12], while Hg accelerates carotid atherosclerosis [6,13].

Increasing evidence shows that vitamin intake could reverse CVDs, diabetes, and mental illness [14], and curry rice (the distinct yellow color of curry rice is mostly derived from a polyphenolic compound, curcumin [1,7-bis(4-hydroxy-3-methoxyphenyl)-1,6-heptadien-3,5-dione]) has been shown to prevent and to be useful for treating CVDs, due to its antioxidant and anti-inflammatory properties [15]. In this study, we consider whether increased intake of vitamins and curry reduces the risk of MetS in the Korean population with or without various NCDs, and whether serum heavy metals and hs-CRP are positively associated with the risk of MetS.

## 2. Materials and Methods

### 2.1. Study Population

We used data obtained by KNHANES studies conducted by the Korean Ministry of Health and Welfare, specifically KNHANES IV (2009), KNHANES V (2010–2012), KNHANES VI (2013), and the KNHANES VII (2016–2017) [16]. These studies were conducted using a stratified, multi-stage, cluster-sampling method and involved consideration of geographic zone, level of urbanization, economic development, gender, and age distribution. Subjects surveyed were randomly selected from 10,533 households (2009), 8958 (2010), 8518 (2011), 8058 (2012), 8018 (2013), 8150 (2016), and 8127 (2017). In the present study, subjects that participated in a health interview and nutrition survey and underwent a health examination survey with adequate information on MetS were selected. Of the 60,362 individuals that participated in a KNHANES survey from 2009 to 2013 and 2016 to 2017, 106 were excluded for missing MetS information, and thus, 60,256 were eligible for data analysis. Written informed consent was required from patients and family members, and informed parental consent was obtained on behalf of all minors before examinations, which were performed by the Health and Nutrition Examination Department of the Korea Centers for Disease Control and Prevention (KCDC). A detailed description of the objectives and methods used and of the legal approvals obtained can be found on the KNHANES website (http://knhanes.cdc.go.kr/, accessed on 17 December 2020). This study was approved by the KNHANES inquiry commission and the Institutional Review Board (IRB) of Sunchon National University and followed the guidelines set out in the Declaration of Helsinki. From 2007 to 2013, KNHANES surveys were conducted with the approval of the IRB of the KCDC (2009-01CON-03-2C, 2010-02CON-21-C, 2011-02CON-06-C, 2012-01EXP-01-2C, 2013-07CON-03-4C, 2013-12EXP-03-5C). However, in 2016 and 2017, KNHANES was conducted without such approval in accord with the opinion of the IRB of the KCDC [17].

### 2.2. Parameters

Information on sociodemographic characteristics, lifestyle, current medications, medical, and family history was collected during health interviews. Alcohol intakes were classified as low or high (high-risk drinking was defined as >5 drinks per day for ≥1 month). Subjects with a lifetime history of smoking of >100 cigarettes and who still smoked daily or occasionally were classified as current smokers; others were classified as ex-/non-smokers.

The physical activity was measured using the modified Global Physical Activity Questionnaire (GPAQ) developed by the WHO, which contains the groups of resistance exercise and weekly walking added to the original GPAQ. Physical activity was dichotomized as regular or irregular. Regular physical activity was defined as: (1) participation in vigorous, ≥20 min per session for ≥3 days a week; (2) or participation in moderate, ≥30 min per session for ≥5 days per week, (3) or participation in walking; ≥30 min per session for ≥5 days a week [18].

Dyslipidemia was defined as the presence of at least one of the following: LDL-C ≥ 160 mg/dL, triglyceride ≥ 200 mg/dL, HDL-C < 40 mg/dL. Hypertension was defined as a systolic blood pressure (SBP) of ≥140 mmHg or a diastolic blood pressure of ≥90 mmHg or being on anti-hypertensive medication. Type 2 diabetes mellitus was defined as a fasting plasma glucose of ≥126 mg/dL, receipt of anti-diabetic medication, or a HbA1c of ≥6.5%. Stroke, angina, myocardial infarction (MI), MI or angina, and other conditions were defined based on physicians’ diagnoses and the presence of these conditions. Depression was also defined based on physicians’ diagnoses, the presence or treatment for depression, or experience of depression during the previous year or of despair to the point that it disturbed daily routine for ≥2 consecutive weeks [19]. Comorbidities were defined as any disease, such as CVDs, hypertension, hyperlipidemia, type 2 diabetes, cancers, thyroid, kidney, arthritis, osteoarthritis, rheumatoid arthritis, or depression, accompanying MetS. A family history of cardiovascular disease was defined as having at least one parent or sibling with a diagnosis of hypertension, ischemic heart disease, or stroke. A family history of type 2 diabetes or hyperlipidemia was defined as having at least one parent or sibling with a diagnosis of type 2 diabetes or hyperlipidemia.

### 2.3. Laboratory Measurements

Height, weight, waist circumference, and blood pressure were measured during medical checkups using standard procedures. BMI (kg/m^2^) was calculated by dividing weight (kg) by height^2^ (m^2^). Waist circumstance (cm) was measured at the midpoint between the bottom of the rib cage and the iliac crest on the mid-axillary line when exhaling. Blood pressures were measured three times with 5-min intervals using a mercury sphygmomanometer with subjects seated after a 5-min stabilization period. Blood pressures used in the analysis were the averages of second and third measurements. Blood samples were collected after ≥ an 8 h fast and analyzed at the Neodin Medical Institute in Korea. An enzymatic assay was used to determine total cholesterol, high-density lipoprotein cholesterol (HDL-C), triglyceride, low-density lipoprotein cholesterol (LDL-C), and fasting blood glucose levels using a Hitachi automated analyzer 7600 (Hitachi, Japan). The HbA1c level was measured using high-performance liquid chromatography (HLC-723G7; Tosoh, Tokyo, Japan). Serum highly sensitive C-reactive Protein (hs-CRP) was determined using Cobas (Cobas; Roche, Germany) and a Cardiac C-reactive Protein Highly Sensitive kit (Cardiac C-reactive Protein Highly Sensitive; Roche, Germany). The reference normal range for hs-CRP is 0 to 3 mg/L [20].

### 2.4. Determination of Serum Pb, Hg, and Cd Levels

Pb, Hg, and Cd analyses were performed as previously described [9]. In brief, these tests were performed by the Neodin Medical Institute, which had been approved by the Korean Ministry of Labor for Heavy Metal Analysis. Furthermore, the tests used met the requirements of the Korea Occupational Safety and Health Administration, the German External Quality Assessment Scheme, and the U.S. CDC. Pb and Cd levels were measured by graphite furnace atomic absorption spectrometry (model AAnalyst 600; Perkin Elmer, Turku, Finland) using Zeeman background correction, and total Hg levels were determined using a direct mercury analyzer (model DMA-80 Analyzer; Bergamo, Italy) and the gold amalgam method (KCDC 2013). Limits of detection (LODs) were 0.223 μg/dL, 0.05 μg/L, and 0.087 μg/L for Pb, Hg, and Cd, respectively. Commercial standard reference materials purchased from Bio-Rad were used for internal quality assurance and control (Lyphocheck Whole Blood Metals Control; Bio-Rad, Hercules, CA, USA).

### 2.5. Urinary Cotinine and Smoking Verification

Spot urinary samples were collected to determine urinary cotinine levels by gas chromatography/mass spectrometry using a PerkinElmer Clarus 600T unit with a detection limit of 1.26 ng/mL. Standard reference materials were used for internal quality assurance and control purposes (ClinChek, RECIPE, Munich, Germany). The G-EQUAS uses a standard protocol to measure urinary cotinine. Subjects with an urinary cotinine level of ≥50 ng/mL were defined as cotinine-verified smokers [21,22].

### 2.6. Metabolic Syndrome

MetS was defined as described by the American Heart Association/National Heart, Lung, and Blood Institute for clinical diagnosis and included abdominal obesity, elevated triglycerides, increased waist circumference, decreased HDL, elevated blood pressure, and elevated plasma glucose [23]. Participants with three or more of the following five risk factors were defined as having MetS: (1) elevated waist circumference (WC ≥80 cm in women and WC ≥ 90 cm in men), (2) elevated triglycerides (TG ≥ 150 mg/dL or receipt of medication for elevated triglycerides), (3) low high-density lipoprotein cholesterol (HDL-C < 50 mg/dL in women, HDL-C < 40 mg/dL in men or receipt of medication to increase HDL-C), (4) elevated blood pressure (systolic blood pressure ≥ 130 mmHg and/or ≥85 mmHg diastolic blood pressure or on antihypertensive drug treatment and a history of hypertension), and (5) elevated fasting glucose (≥100 mg/dL or receipt of medical treatment for elevated glucose) [23,24,25].

### 2.7. Serum Levels of Vitamins A, D, and E

Serum 25-hydroxyvitamin D (25(OH)D) levels were determined using a 1470 Wizard gamma counter (Perkin Elmer, Turku, Finland) and a radioimmunoassay (RIA) (DiaSorin, Stillwater, MN, USA). Vitamin D deficiency was defined as a serum 25 (OH)D concentration of <50 nmol/L. Serum retinol (mg/L) and serum α-tocopherol (mg/L) levels were measured with an Agilent1200 (Agilent1200; Agilent, Santa Clara, CA, USA) using Chromsystems (Chromsystems; Chromsystems Instruments & Chemicals, Munich, Germany) reagents. The reference normal ranges for serum retinol and α-tocopherol in adults are 0.30 to 0.70 mg/L and 5.00 to 20.00 mg/L, respectively [20].

### 2.8. Food and Vitamin Intakes

Daily food intakes were calculated using the one-day 24 h recall method. Before food intakes were evaluated, all participants were instructed to maintain their normal dietary habits. Dietary intake information was also collected by administering a semi-quantitative questionnaire on food frequency, which addressed the intakes of 63 food products. Dietary intake information was completed by each participant. Food consumptions were calculated using nine categories: “never or rarely”, “once a month”, “two to three times a month”, “one to two times a week”, “three to four times a week”, “five to six times a week”, “once a day”, “twice a day”, and “three or more times every day [26]”.

Three food types (green vegetables, other vegetables, and fruit) were chosen from the 63 food items. Green vegetables included spinach, cucumber, radish leaves, and pepper; other vegetables included radish, sprouts, Korean cabbage, cabbage, pumpkin, carrot, and tomato; and fruits included persimmon, tangerine, pear, watermelon, grape, peach, strawberry, apple, banana, and citrus. Subjects were divided into two groups for each food type based on food intake frequencies—that is, to low consumption (“almost never”, or “less than three times a month”) or high consumption groups (“2–6 times per week”) [18].

The curry consumption was estimated using responses to the KNHANES food frequency questionnaire. Curried rice was the only food among the surveyed foods related to curry consumption. Subjects were divided into a low curry consumption group (“almost never”, or “less than three times a month”) or a high curry consumption group (“2–6 times per week”) based on curry consumption frequency [27].

Daily vitamin intake was determined by summing mean 24 h dietary intakes using Can-Pro 3.0 nutrient intake assessment software developed by the Korean Nutrition Society. Daily total energy intakes were calculated using Korean Estimated Energy Requirements (EERs) [28]. Total vitamin A intake was measured by summing vitamin A and b-carotene intakes and dividing by 6 [18].

### 2.9. Statistical Analysis

The statistical analysis was performed using STATA software (version 16.0; StataCorp, College Station, TX, USA). The baseline characteristics of participants were summarized using frequencies and proportions for categorical variables, and means and standard deviations for continuous variables. Continuous and categorical variables were analyzed using the Student’s *t*-test and the *χ*^2^ test, respectively.

Associations between MetS and related factors were examined by logistic regression. First, potential covariates were identified in the literature, based on subjective prior knowledge, or by *p* values ≤ 0.25 by univariate analysis, and were entered into the full model [29]. In the multivariate analysis, logistic regression models were used to identify risk factors associated with MetS; these included vitamins (mg or μg), heavy metals (μg/L), curry consumption (low or high consumption), age (≤29, 30–39, 40–49, 50–59, ≥60), sex, residential area (rural vs. urban), marital status (married, living alone), education level (≤middle school, high school, ≥college), monthly household income (<2000, ≥2000 and <4000, ≥4000 and <6000, ≥6000), smoking status (current smoker, non/ex-smoker), cotinine-verified smoker (yes, no), high-risk drinking (yes, no), physical activity (not regular, regular), BMI (<18.5, ≥18.5 and <25, ≥25 and <30, ≥30), green vegetables (low or high consumption), other vegetables (low or high consumption), fruit (low or high consumption), and comorbidities (yes, no) (detailed in Appendix A). To visualize the moderating effect of the MetS, marginal effects were performed using the results of logistic regression analysis. Forest plots were also produced from the models to visualize associations between MetS and risk factors. Statistical tests were two-sided, and *p*-values < 0.05 were considered statistically significant.

## 3. Results

### 3.1. Characteristics of Participants with Respect to Metabolic Syndrome

We included 60,256 subjects that participated in the KNANES 2009–2103 and 2016–2017 surveys. Mean participant age was 40.8 ± 22.8 years (min–max: 15–80), and 32,827 (54.5%) were women. Most subjects reported that they never or rarely consumed curry (53%), 46% reported that they consumed curry occasionally (“2–3 times a month” or “once a week”), while only 1% (67/10,874) of the subjects reported that they consumed curry often (“2–4 times a week” or “5–6 times a week”). Among the subjects with MetS (*n* = 21,612), there were significantly more females than males and more were aged ≥ 60, unemployed, living in a rural location or alone, had a low education level, a low monthly income household, were underweight (BMI < 18.5 Kg/m^2^), overweight (BMI ≥ 25 and <30 Kg/m^2^), obese (BMI ≥ 30 Kg/m^2^), or had a family history of CVD or diabetes. Notably, the risk of MetS was significantly higher for subjects with comorbidities among those without low curry, other vegetable, or fruit consumptions. The characteristics of the study population by MetS are shown in Table 1.

### 3.2. Characteristics of Daily Vitamin Intakes and Serum Heavy Metal Levels

Average daily intakes of vitamin B1, B2, B3, and C were 1.35 ± 0.86 mg (95% CI 1.34–1.36); 1.28 ± 0.82 mg (95% CI 1.28–1.29); 14.64 ± 9.08 mg (95% CI 14.56–14.72); and 90.10 ± 95.76 mg (95% CI 89.30–90.90), respectively. Average daily intakes of total vitamin A (mean ± SE) and retinol were 692.12 ± 4.44 μg (95% CI 683.43–700.81) and 125.36 ± 345.14 μg (95% CI 122.44–128.28), respectively. Average levels of serum Pb, Hg, Cd, and hs-CRP were 2.06 ± 1.10 μg/dL (95% CI 2.04–2.07); 4.08 ± 3.53 μg/L (95% CI 4.02–4.13); 1.02 ± 0.67 μg/L (95% CI 1.01–1.03), and 1.20 ± 2.02 mg/L (95% CI 1.17–1.24), respectively. No sample had a value of below an LOD. Serum vitamin A, E, and D concentrations were 0.54 ± 0.19 mg/L (95% CI 0.53–0.55), 13.72 ± 6.35 mg/L (95% CI 13.43–14.00), and 17.54± 6.16 nmol/L (95% CI 17.47–17.60), respectively. Vitamin B1 (1.24 ± 0.82 vs. 1.44 ± 0.87, *p* < 0.001), B2 (1.17 ± 0.76 vs. 1.36 ± 0.84, *p* < 0.001), B3 (13.08 ± 8.61 vs. 15.81 ± 9.25, *p* < 0.001), C (83.03 ± 87.46 vs. 95.38 ± 101.23, *p* < 0.001), and A (628.67 ± 6.78 vs. 740.24 ± 5.86, *p* < 0.001) intakes were significantly lower in subjects with than without MetS. On the other hand, serum levels of Pb (2.34 ± 1.22 vs. 1.96 ± 1.03, *p* < 0.001), Hg (4.78 ± 3.87 vs. 3.83 ± 3.37, *p* < 0.001), Cd (1.26 ± 0.69 vs. 0.93 ± 0.64, *p* < 0.001), vitamin A (0.61 ± 0.21 vs. 0.51 ± 0.18, *p* < 0.001), E (15.78 ± 8.33 vs. 12.95 ± 5.23, *p* < 0.001), and hs-CRP (1.58 ± 2.33 vs. 1.02 ± 1.83, *p* < 0.001) were significantly higher in subjects with MetS.

After adjustment for comorbidities, the risk of MetS was found to be significantly lower for subjects with high curry consumption than subjects with low curry consumption (OR 0.64; 95% CI, 0.58—0.70, *p* < 0.001). Furthermore, risks of MetS were reduced by 44% (OR 0.56; 95% CI, 0.54—0.58, *p* < 0.001), 38% (OR 0.62; 95% CI, 0.61—0.64, *p* < 0.001), 7% (OR 0.93; 95% CI, 0.92—0.93, *p* < 0.001), and 1% (OR 0.99; 95% CI, 0.98—0.99, *p* < 0.001), when vitamin B1, B2, B3, and C intakes increased by one mg daily, respectively. However, risks of MetS were increased by 19% (OR 1.19; 95% CI, 1.15—1.24, *p* < 0.001), 3% (OR 1.03; 95% CI, 1.02—1.05, *p* < 0.001), 113% (OR 2.13; 95% CI, 2.00—2.27, *p* < 0.001), and 6% (OR 1.06; 95% CI, 1.03—1.09, *p* < 0.001), when serum levels of Pb, Hg, Cd, and hs-CRP increased by one unit.

After adjustment for potential confounders including monthly household income, residential area, energy intake, occupation, sex, family history of CVDs, family history of diabetes mellites, family history of hyperlipidemia, BMI, smoking status, cotinine verified smoker, high-risk drinking, physical activity, education level, hypertension, dyslipidemia, type 2 diabetes, stroke, myocardial infarction or angina, myocardial infarction, angina, arthritis, osteoarthritis, rheumatoid arthritis, kidney failure, depression, thyroid disease, asthma, and green vegetable, other vegetable, and fruit consumption (details are shown in Appendix A), adjusted odds ratios followed a similar pattern. The risk of MetS was significantly lower for high than low curry consumers (OR 0.85; 95% CI, 0.74—0.98, *p* = 0.028) and reduced by 12% (OR 0.88; 95% CI, 0.79—0.97, *p* = 0.013) and 1% (OR 0.99; 95% CI, 0.98—0.99, *p* = 0.032) when vitamin B1 and C intakes increased by one mg, respectively. Similarly, risks of MetS were increased by 14% (OR 1.14; 95% CI, 1.03—1.28, *p* = 0.015), 3% (OR 1.03; 95% CI, 1.02—1.06, *p* = 0.031), and 9% (OR 1.09; 95% CI, 1.04—1.14, *p* = 0.001), when serum Pb, Hg, and hs-CRP levels increased by one unit. Crude odds ratios and adjusted odds ratios (95% confidence interval) for risks of MetS are shown in Figure 1.

Figure 2 shows the marginal effects of vitamin intakes, curry consumption, serum heavy metals, and hs-CRP on MetS by age group in the Korean population after adjustment for potential cofounders. The effects of vitamin and curry intakes showed similar trends. The probability of MetS decreased when vitamin or curry intakes increased but increased when serum Pb, Hg, or hs-CRP levels increased.

## 4. Discussion

Our epidemiological findings provide evidence that supports experimental knowledge regarding associations between vitamin intake, curry consumption, and heavy metal exposure and MetS and subjects with MetS and comorbidities. We identified associations between MetS and vitamin intakes, serum heavy metal levels, hs-CRP, and comorbidities in a cohort representative of the Korean population. Vitamin intakes and high curry consumption exhibited inverse correlations with the prevalence of MetS, while serum levels of heavy metals and hs-CRP were positively correlated with MetS.

Dramatic global increases in urbanization and industrialization have increased the risk of exposure to pollutants, especially heavy metals [30]. The growing global burden posed by NCDs has made their prevention and management a priority, and this is especially true in the context of the COVID-19 pandemic because COVID-19 is associated with NCD-induced morbidity and mortality [31].

In this study, we found that levels of serum heavy metals and cardiometabolic risk factors were positively correlated with MetS, which agrees with the results of a previous study [9,32]. Heavy metals such as Pb, Hg, and Cd are toxic and can trigger various diseases, especially CVDs [33]. In addition, they can increase ROS and reactive nitrogen species levels and cause oxidative stress, DNA damage, and the oxidation of protein thiol groups [34]. Furthermore, heavy metals provoke the production of inflammatory cytokines and anti-thrombotic agents [6,7]. Recent data also indicate that elevated hs-CRP levels increase the risk of MetS development in obese and non-obese women [35]. These findings indicate that the harmful impacts of multiple environmental pollutants, including heavy metals, on MetS should be urgently addressed and that a preventative strategy targeting the high-risk population is required to reduce the negative impacts of environmental pollutants and heavy metals.

Vitamin B1 plays an important role in intracellular glucose metabolism by acting as a coenzyme for α-ketoglutarate dehydrogenase complexes, transketolase, and pyruvate dehydrogenase [36], and our study shows that vitamin B1 intakes were significantly lower and that levels of HbA1c and fasting glucose were significantly higher in individuals with MetS. It has been shown that reduced vitamin B1 levels in diabetic vascular cells exacerbate metabolic dysfunction in hyperglycemia [37], and genetic studies have reported that the relationship between diabetes and vitamin B1 is associated with transketolase (T*k*), α-1-antitrypsin, the *SLC19A2* gene, and p53 [38,39,40,41]. Vitamin B1 and its derivatives may hinder the biochemical pathways leading to caspase activation, for example, by increased flux via the polyol or hexosamine biosynthesis pathways, inducing the production of advanced glycation end-products or activation of protein kinase C [42,43,44,45]. Our findings are in line with those of a previous study, in which thiamine at 150 mg daily for 1 month significantly reduced plasma fasting glucose concentrations in patients with drug-naïve type 2 diabetes [46]. The consumption of vitamin B1 has also been reported to be inversely related to dyslipidemia [47]. Vitamin B1 attenuates the adverse consequences of high endothelial glucose levels by reducing the glycation of intracellular proteins [48], plays a vital role in the prevention of atherosclerotic plaque, and inhibits the glucose and insulin-mediated proliferation of human infragenicular arterial smooth muscle cells [49]. Several studies have shown that regular vitamin B1 administration increases the functions of endothelial cells and retards the development of atherosclerosis [50], and short-term vitamin B1 treatment regenerated endothelial cell functions in otherwise healthy smokers with endothelial dysfunction caused by smoking [51].

Oxidative stress could play a significant part in the etiology of MetS [52], and MetS patients exhibit systemic oxidative damage due to ROS upregulation and/or reduced levels of antioxidant enzymes [53]. Although our findings show that regular vitamin B1 and C intake reduces the risk of MetS, other studies have reported associations between vitamin B2, B3, A, E, and MetS [18,54,55]. It has been reported that vitamin B2 prevents pro-inflammatory activity in adipocyte and macrophage co-cultures and thus might reduce the mild inflammation associated with obesity [54]. Another study showed that vitamin B2 deficiency might increase the pro-inflammatory activities of adipocyte cells and result in chronic inflammation in the obese [56]. Vitamin B3 has also been found to be an efficient antioxidant that reduces ROS production and prevents DNA damage in lymphocytes [57], which is in line with the observation that vitamin B3 treatment promoted the normalization of low HDL-C atherogenic dyslipidemia [58]. Furthermore, a negative correlation was reported between diets enriched with specific antioxidants, such as vitamin C, A, and E, and oxidative stress [59]. These findings show that efforts are required to establish targeted vitamin B1 and C intakes in Korea. We believe these strategies would effectively diminish the prevalence of MetS.

Our analyses revealed that the risk of MetS was significantly lower among high curry consumers, which supports our hypothesis that high curry consumption reduces the risk of type 2 diabetes and the findings of previous studies [60]. Curcumin could improve endothelial function and reduce oxidative stress and levels of inflammatory markers (IL-6, TNF alpha, endothelin-1) in type 2 diabetes patients and enhance the functions of β cells. [61,62]. Of note, curcumin also has an impact on insulin secretion in healthy subjects [63], and in patients with acute coronary syndrome, curcumin reduced triglyceride, total cholesterol, and LDL-C levels but increased HDL-C in [64]. Furthermore, recent data indicate that curcumin has a preventive effect on stroke by reducing oxidative stress related to signaling the uncoupling of protein 2, thus strengthening endothelial vascular function [65]. Curcumin also has a profound effect on microglial response, facilitates microglial M2 polarization, and prevents pro-inflammatory responses by microglia. Additionally, in ischemic stroke patients, curcumin post-treatment diminished brain damage and strengthened vascular endothelial function [66]. These data demonstrate the potential benefits of curcumin for diabetes and CVDs and show that curcumin is not only a promising alternative therapeutic for type 2 diabetes but also offers a preventive strategy for CVDs.

To the best of our knowledge, this large-scale study is the first to report the combined effects of vitamins, curry consumption, and serum heavy metal and hs-CRP levels on MetS at a national level in Korea. However, the study has several limitations. First, the cross-sectional design of the KNHANES studies prevented our evaluation of causality between MetS and these factors. Second, as no physiological markers of antioxidant status were measured, oxidation status and vitamin levels in serum (such as B vitamins, vitamin C), vitamin E intakes of food source, and tissues were not evaluated. Third, consumptions of food and vitamins were measured using one-day 24 h recall data and thus may have been under- or overestimated. However, all participants were instructed to maintain their usual dietary habits and the one-day 24 h recall method is commonly used to assess food intake. Finally, relatively few individuals consumed curry frequently, which may have influenced results regarding the association between curry consumption and MetS.

## 5. Conclusions

The prevalence of MetS and heavy metal exposure in Korea show increasing trends [67,68,69], and these have worsened during the COVID-19 pandemic [70]. MetS is a significant risk factor of severe COVID-19, but the mechanisms involved have yet to be determined. Dramatic increases in urbanization and industrialization have elevated risks of heavy metal exposure, and it is critical that a prevention strategy targeting the high-risk population be devised to reverse this trend. Vitamins and curry supplements might reduce the prevalence of MetS, and our results indicate that thiamine and curry intakes might protect the public against the dual burden of communicable and non-communicable diseases in Korea. However, further work is required to determine the nature of the mechanisms involved.

## Figures and Tables

**Figure 1 antioxidants-10-00808-f001:**
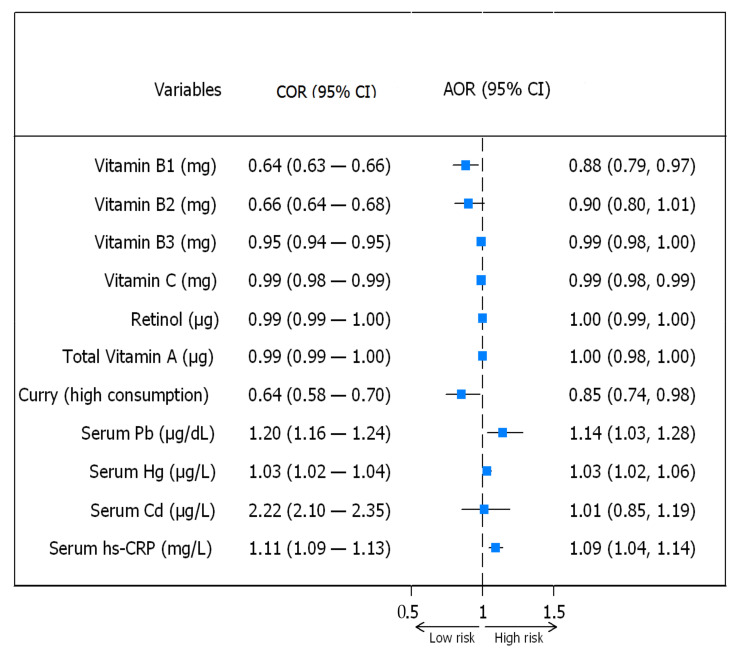
Crude odds ratio and adjusted odds ratio (95% confidence interval) for risks of MetS.

**Figure 2 antioxidants-10-00808-f002:**
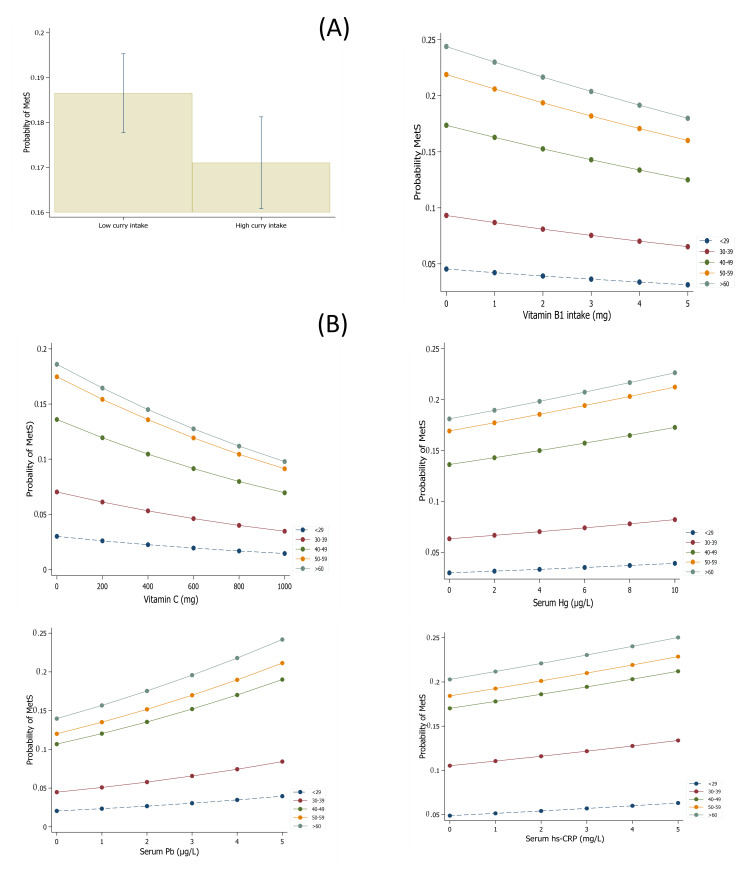
Marginal effects of vitamin intakes, curry consumption (**A**), and serum levels of heavy metals and hs-CRP (**B**) on MetS by age group in the Korean population after adjustment for potential cofounders.

**Table 1 antioxidants-10-00808-t001:** Demographic distribution of participants by metabolic syndrome.

Variables	No.	Metabolic Syndrome		ORs 95%CI	*p*-Value
		Yes	No		
Sex (%)	60,256				
Male	27,429	8239 (38.1)	19,190 (49.7)	1 (refer)	
Female	32,827	13,373 (61.9)	19,454 (50.3)	1.60 (1.55–1.66)	<0.001
Age (year) ^†^	60,256	39.6 ± 28.0	41.6 ± 19.1		<0.001
Age group (%)					
<29	19,626	8353 (38.7)	11,273 (29.2)	1 (refer)	
30–39	8332	1219 (5.6)	7113 (18.4)	0.23 (0.22–0.25)	<0.001
40–49	8656	1946 (9.0)	6710 (17.4)	0.39 (0.37–0.41)	<0.001
50–59	8554	2861 (13.2)	5693 (14.7)	0.68 (0.64–0.72)	<0.001
>60	15,088	7233 (33.5)	7855 (20.3)	1.24 (1.19–1.30)	<0.001
Marital status (%)	60,126				
Married	39,286	12,929 (60.0)	26,357 (68.3)	1 (refer)	
Living alone	20,840	8611 (40.0)	12,229 (31.7)	1.44 (1.39–1.49)	<0.001
Residential areas (%)	60,256				
Urban	48,396	16,919 (78.3)	31,477 (81.4)	1 (refer)	
Rural	11,860	4693 (21.7)	7167 (18.6)	1.22 (1.17–1.27)	<0.001
Occupation (%)	44,687				
Managers, professional	5358	621 (5.6)	4737 (14.1)	1 (refer)	
Office worker, clerical workers	3790	497 (4.5)	3293 (9.8)	1.15 (1.01–1.31)	0.029
Service workers, sales workers	5407	1217 (11.0)	4190 (12.5)	2.22 (1.99–2.46)	<0.001
Agriculture, forestry, and fishing workers	2848	900 (8.1)	1948 (5.8)	3.52 (3.14–3.95)	<0.001
Craft, plant, and machine operators and assemblers	4029	677 (6.1)	3352 (10.0)	1.54 (1.37–1.73)	<0.001
Elementary occupations	3730	1111 (10.1)	2619 (7.7)	3.24 (2.90–3.61)	<0.001
Unemployed	19,525	6036 (54.6)	13,489 (40.1)	3.41 (3.12–3.73)	<0.001
Education level (%)	55,326				
≤Middle school	27,702	13,669 (76.4)	14,033 (37.5)	1 (refer)	
High school	14,342	2689 (15.0)	11,653 (31.1)	0.24 (0.23–0.25)	<0.001
≥College	13,282	1544 (8.6)	11,738 (21.4)	0.14 (0.13–0.14)	<0.001
Monthly household income (%) ^§^	59,628				
<2000	16,917	7264 (34.0)	9653 (25.2)	1 (refer)	
≥2000 and <4000	19,423	6922 (32.4)	12,501 (32.7)	0.74 (0.71–0.77)	<0.001
≥4000 and <6000	13,065	4176 (19.6)	8889 (23.2)	0.62 (0.60–0.65)	<0.001
≥6000	10,223	2982 (14.0)	7241 (18.9)	0.55 (0.52–0.58)	<0.001
BMI group (%)	56,009				
<18.5	8945	5057 (29.0)	3888 (10.1)	5.71 (5.43–6.01)	<0.001
≥18.5 and <25	31,942	5925 (33.9)	26,017 (67.5)	1 (refer)	
≥25 and <30	13,096	5337 (30.6)	7759 (20.1)	3.02 (2.89–3.16)	<0.001
≥30	2026	1137 (6.5)	889 (2.3)	5.62 (5.12–6.16)	<0.001
Curry consumption (%)	10,874				
Rarely or never	5812	1587 (61.9)	4225 (50.8)	1 (refer)	
Often and occasionally	5062	977 (38.1)	4085 (49.2)	0.64 (0.58–0.70)	<0.001
Green vegetable consumption (%)	10,874				
Rarely or never	5838	1419 (55.3)	4419 (53.2)	1 (refer)	
Often and occasionally	5036	1145 (44.7)	3891 (46.8)	0.92 (0.84–1.00)	0.055
Other vegetable consumption (%)	10,914				
Rarely or never	7468	1809 (70.2)	5659 (67.9)	1 (refer)	
Often and occasionally	3446	769 (29.8)	2677 (32.1)	0.90 (0.82–0.99)	0.029
Fruit (%)	10,916				
Rarely or never	6691	1726 (66.9)	4965 (59.5)	1 (refer)	
Often and occasionally	4225	852 (33.1)	3373 (30.5)	0.73 (0.66–0.80)	<0.001
Cotinine verified smoker (%)	60,256				
No	16,780	4468 (20.7)	12,312 (31.9)	1 (refer)	
Yes	43,476	17,144 (79.3)	26,332 (68.1)	1.79 (1.73–1.89)	<0.001
Smoking status (%)	42,803				
Non-/ex-smoker	32,992	9413 (84.5)	23,579 (74.5)	1 (refer)	
Current smoker	9811	1728 (15.5)	8083 (25.5)	0.54 (0.51–0.57)	<0.001
Drinking status (%)	47,435				
Often	15,884	4814 (42.8)	11,070 (30.6)	1 (refer)	
Occasionally	22,403	4511 (40.1)	17,892 (49.4)	0.58 (0.55–0.61)	<0.001
Never or rarely	9148	1915 (17.0)	7233 (20.0)	0.61 (0.57–0.65)	<0.001
Physical activity (%)	60,256				
Not regular	51,088	19,677 (91.1)	31,411 (81.3)	1 (refer)	
Regular	9168	1935 (8.9)	7233 (18.7)	0.43 (0.40–0.45)	<0.001
Family history of CVDs (%)	60,256				
No	51,088	4337 (50.0)	17,727 (60.2)	1 (refer)	
Yes	9168	4331 (50.0)	11,703 (39.8)	1.51 (1.44–1.59)	<0.001
Family history of diabetes (%)	37,720				
No	30,335	6375 (75.5)	23,960 (81.8)	1 (refer)	
Yes	7385	2071 (24.5)	5314 (18.2)	1.46 (1.38–1.55)	<0.001
Family history of hyperlipidemia (%)	36,414				
No	34,145	7525 (94.2)	26,620 (93.6)	1 (refer)	
Yes	2269	461 (5.8)	1808 (6.4)	0.90 (0.81–1.00)	0.055
Comorbidities ^¶^					
Type 2 diabetes mellitus	3793	2525 (13.9)	1268 (3.3)	4.67 (4.35–5.01)	<0.001
Hypertension	9837	5927 (32.6)	3910 (10.3)	4.21 (4.03–4.41)	<0.001
Dyslipidemia	5532	3332 (32.8)	2200 (6.0)	7.59 (7.15–8.06)	<0.001
Stroke	906	486 (2.7)	420 (1.12)	2.45 (2.15–2.80)	<0.001
MI or angina	1130	565 (3.4)	565 (1.6)	2.21 (1.96–2.48)	<0.001
MI	393	178 (1.0)	215 (0.6)	1.73 (1.42–2.12)	<0.001
Angina	810	421 (2.3)	389 (1.0)	2.29 (1.99–2.63)	<0.001
Asthma	1965	804 (4.5)	1161 (3.1)	1.47 (1.34–1.61)	<0.001
Thyroid disease	1607	585 (3.3)	1022 (2.7)	1.20 (1.08–1.33)	0.001
Osteoarthritis	4984	2706 (15.1)	2278 (6.1)	2.74 (2.58–2.91)	<0.001
Rheumatoid arthritis	880	378 (2.1)	502 (1.3)	1.58 (1.38–1.81)	<0.001
Arthritis	5628	2948 (17.5)	2680 (7.3)	2.68 (2.53–2.83)	<0.001
Kidney failure	178	89 (0.5)	89 (0.2)	2.09 (1.56–2.81)	<0.001
Depression	1785	706 (3.9)	1079 (2.9)	1.38 (1.25–1.52)	<0.001
Waist circumference (cm) ^†^	56,935	77.6 ± 10.5	74.2 ± 19.0	--	<0.001
Total cholesterol (mg/dL) ^†^	47,054	195.9 ± 40.9	183.8 ± 35.6	--	<0.001
LDL-C (mg/dL) ^†^	10,339	117.4 ± 36.3	111.3 ± 31.9	--	<0.001
Triglyceride (mg/dL)	47,054	194.7 ± 136.4	110.9 ± 87.7	--	<0.001
HDL-C (mg/dL) ^†^	47,054	44.3 ± 9.8	51.6 ± 12.0	--	<0.001
HbA1c (%) ^†^	33,118	6.4 ± 1.2	5.6 ± 0.7	--	<0.001
Fasting glucose (mg/dL) ^†^	46,984	113.7 ± 32.0	93.8 ± 16.6	--	<0.001
Energy intake (Kcal) ^†^	53,701	1680.9 ± 797.4	2026.6 ± 880.2	--	<0.001
Hemoglobin (g/dL) ^†^	46,846	13.7 ± 1.5	14.0 ± 1.6	--	<0.001
ALT (U/L) ^†^	47,054	24.9 ± 17.9	19.7 ± 17.7	--	<0.001
AST (U/L) ^†^	47,054	24.8 ± 12.9	21.3 ± 13.1	--	<0.001
Systolic blood pressure (mmHg) ^†^	50,220	130.3 ± 17.4	114.4 ± 15.2	--	<0.001
Diastolic blood pressure (mmHg)	50,220	78.8 ± 11.4	73.6 ± 10.4	--	<0.001

^†^: means ± SDs, two-sample *t*-test with unequal variances, except for ALT with equal variances; ^¶^: reference with no comorbidities; ^§^: thousand won. AST: aspartate aminotransferase. ALT: alanine aminotransferase, MI: myocardial infarction. BMI: body mass index (kg/m^2^), CVD: cardiovascular disease, HDL: high-density lipoprotein, LDL-C: low-density lipoprotein cholesterol.

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
