# Peer review of "Effects of Antioxidant Vitamins, Curry Consumption, and Heavy Metal Levels on Metabolic Syndrome with Comorbidities: A Korean Community-Based Cross-Sectional Study"

_antioxidants, 2021, doi:10.3390/antiox10050808_

Round 1

Reviewer 1 Report

This paper deals with the relationship between dietary factors (vitamins and curry intake) and the risk of metabolic syndrome in Korea. In addition, the potential effect of the exposure to HMs (measured in serum) on the risk of metabolic syndrome was studied as well. Although the subject is very interesting, the manuscript needs major revisions. My comments are below.

  1. English should be improved (especially in Introduction and Discussion sections).
  2. Too many long sentences make the manuscript hard to read and understand e.g. the sentence in lines 43-45 “Remarkedly …” or lines 290-292 “Our findings …” or lines 301-302 “Because the COVID-19 pandemic …” or lines 324-327 “Vitamin B1 and ….”.
  3. Line 48: the authors state that Pb and Cd increase the risk of CVD. However they do not include any relevant reference to support this statement.
  4. Many sentences in discussion section sound unclear e.g line 296 “these findings …”, line 327 “these data support …”
  5. Line 319 “…the levels of vitamin intake..” which vitamin did the authors mean?
  6. Lines 343-344 “our findings only show that intake of vitamin C…” what about other vitamins which were examined in this study?
  7. The last paragraph (page 22-23) in the discussion section was duplicated in conclusions.

Author Response

SUNCHON NATIONAL UNIVERSITY

57922 255 Jungang-ro, Suncheon-si,

Jeollanam-do (Seokhyeon-dong), Korea

Tel: 061-750-3132~3 Fax 061-750-3139

E http://www.scnu.ac.kr/ggs/main.do

The Editor

Antioxidant

2021 May 03rd  

RE: Re-submission of a manuscript

Dear Sir/Madam,

On behalf of co-authors, we would like to submit our manuscript entitled, “Effects of antioxidant vitamins, curry consumption and heavy metal levels on the metabolic syndrome with comorbidities: a Korean community-based cross-sectional study” to your journal and request that you consider publishing it as a research article.

In this version, we make a point-by-point response to the comments in the table below. We edited English for the whole manuscript, and we only highlighted in red in our manuscript based on the reviewer’s comments. We hope this version will better than the previous one. Your consideration of our manuscript would be highly appreciated.

Yours sincerely,

Min-Sun Kim, PhD

Department of Pharmacy, College of Pharmacy, Sunchon National University

Sunchon 57922, Republic of Korea

Tel: +821025101635; Email: minsun@scnu.ac.kr

Response to Reviewer 1 Comments

Point 1: This paper deals with the relationship between dietary factors (vitamins and curry intake) and the risk of metabolic syndrome in Korea. In addition, the potential effect of the exposure to HMs (measured in serum) on the risk of metabolic syndrome was studied as well. Although the subject is very interesting, the manuscript needs major revisions. My comments are below.

English should be improved (especially in Introduction and Discussion sections).

Response 1: Thank you for your kind words. First, we attempted to examine the grammatical structure. Following that, we sent this manuscript to a colleague for proofreading. We're hoping that this version is superior to the previous one.

Point 2: Too many long sentences make the manuscript hard to read and understand e.g. the sentence in lines 43-45 “Remarkedly …” or lines 290-292 “Our findings …” or lines 301-302 “Because the COVID-19 pandemic …” or lines 324-327 “Vitamin B1 and ….”.

Response 2: Thanks for your comments. We changed it into:

- Line 43-45: In addition to lifestyle and genetic factors, heavy metals are also risk factors of MetS. Levels of heavy metals, especially lead (Pb), mercury (Hg), and cadmium (Cd), released to the environment by vehicles or factories or in contaminated seafood are increasing and accumulate in bones, kidney cortices, and lungs. Heavy metals catalyze the release of reactive oxygen species (ROS), inflammatory mediators, and antithrombotic substances that damage vascular endothelial cells and exacerbate hypertension. In particular, Pb and Cd disrupt blood clotting and increase the risk of CVD, while Hg accelerates carotid atherosclerosis.

- Line 290-292: Our epidemiological findings provide evidence that supports experimental knowledge regarding associations between vitamin intake, curry consumption, and heavy metal exposure and MetS and subjects with MetS and comorbidities.

- Line 301-302: The growing global burden posed by NCDs has made their prevention and management a prior-ity, and this is especially true in the context of the COVID-19 pandemic because COVID-19 is as-sociated with NCD-induced morbidity and mortality.

- Line 324-327: Vitamin B1 and its derivatives may hinder the biochemical pathways leading to caspase activation, for example, by increased flux via the polyol or hexosamine biosynthesis pathways, inducing the productions of advanced glycation end-products or activation of protein kinase C.

Point 3: Line 48: the authors state that Pb and Cd increase the risk of CVD. However they do not include any relevant reference to support this statement.

Response 3: Thanks for your comments. We added more relevant reference in this version. In particular, Pb and Cd disrupt blood clotting and increase the risk of CVD [7,12].

  1. Angeli, J.K.; Pereira, C.A.C.; de Oliveira Faria, T.; Stefanon, I.; Padilha, A.S.; Vassallo, D.V. Cadmium exposure induces vascular injury due to endothelial oxidative stress: the role of local angiotensin II and COX-2. Free radical biology and medicine 2013, 65, 838-848.
  2. Vaziri, N.D. Mechanisms of lead-induced hypertension and cardiovascular disease. American Journal of Physiology-Heart and Circulatory Physiology 2008, 295, H454-H465.

Point 4: Many sentences in discussion section sound unclear e.g line 296 “these findings …”, line 327 “these data support …”

Response 4: Thanks for your comments. We deleted this sentence in this verion Line 296

We deleted this sentence in this verion Line 327 we changed into “Our findings are in line with those of a previous study, in which thiamine at 150 mg daily for a 1 month significantly reduced plasma fasting glucose concentrations in patients with drug-naïve type 2 diabetes”.

Point 5: Line 319 “…the levels of vitamin intake.” which vitamin did the authors mean?

Response 5: Thanks for your comments. Line 319 we changed into “our study shows vitamin B1 intakes were significantly lower and that levels of HbA1c and fasting glucose were significantly higher in individuals with MetS”.

Point 6: Lines 343-344 “our findings only show that intake of vitamin C…” what about other vitamins which were examined in this study?

Response 6: Thanks for your valuable comments. Line 343-344. We changed into “Although our findings show that regular vitamin B1 and C intake reduce the risk of MetS, other studies have reported associations between vitamin B2, B3, A, E, and MetS”.

Point 7: The last paragraph (page 22-23) in the discussion section was duplicated in conclusions.

Response 7: Thanks for your suggestion. We deleted paragraph line 385-394

We only keep one paragraph in the conclusion. “The prevalence of MetS and heavy metal exposure in Korea show increasing trends, and these have worsened during the COVID-19 pandemic. MetS is a significant risk factor of severe COVID-19, but the mechanisms involved have yet to be determined. Dramatic increases in urbanization and industrialization have elevated risks of heavy metal exposure, and it is critical that a prevention strategy targeting the high-risk population be devised to reverse this trend. Vitamins and curry supplements might reduce the prevalence of MetS, and our results indicate that thiamine and curry supplementation might protect the public against the dual burden of communicable and non-communicable diseases in Korea. However, further work is required to determine the natures of the mechanisms involved”.

Reviewer 2 Report

Effects of antioxidant vitamins, curry consumption and heavy 2 metal levels on the metabolic syndrome with comorbidities: a 3 Korean community-based cross-sectional study

Overall, this is an interesting article on dietary and blood marker risk factors for metabolic syndrome.  It is unique in using blood values for heavy metals as risk factors.  There are several concerns about the value of 3 large figures.  Also, the English needs major help.

Abstract

Lines 13-14: The use of 24-hr recall and food frequency questionnaire are confusing here.  Which is used for what?

Line 15: Change “Heavy metals” to Serum heavy metal”.

There is no mention of the cohort used in this study.  Is it cross-sectional; is it adults; is it Korean?

Methods

Lines 84-87: Physical activity is poorly defined.  It appears that regular physical activity has 3 requirements: vigorous physical activity, moderate physical activity, and walking.  These sum to at least 6 hours per week.  Is this correct?  If this is regular, then what is irregular?  Everybody else with less than 6 hours?

Line 147: 25- hydroxy-Vitamin D should be spelled out.

Results

Line 202: What were the age limits, lowest and highest, on KNANES participants?

Line 207: Underweight is BMI<18.5

Lines 201-216: Since all of these results are shown in Table 1, it is not really necessary to describe them in the text.  You might highlight a few of the more interesting ones, such as curry.  Specifically, you do not need to mention the 5 markers that are part of the MetS.  These are self-evident.

Line 228: The value of 125 ug/day for the average intake of vitamin E is suspicious.  This is well below the average of 8.4 mg/day for middle aged Americans and below a value of 20 mg/day  for total vitamin E and 3 mg/day for alpha-tocopherol from the following reference for middle aged Koreans.

Nutr Res Pract. 2015 Apr;9(2):192-8. doi: 10.4162/nrp.2015.9.2.192. Epub 2015 Mar 4.

“Vitamin E status of 20- to 59-year-old adults living in the Seoul metropolitan area of South Korea Dietary vitamin E and total vitamin E intake (dietary plus supplemental vitamin E) was 17.68 ± 14.34 and 19.55 ± 15.78 mg α-tocopherol equivalents, respectively. The mean daily α-tocopherol, and γ-tocopherol intakes were 3.07 ± 2.27 mg and 5.98 ± 3.74 mg, respectively.”

Line 229: The average blood value for Pb is given as 2.06 ug/L.  In line 122 of the Methods, the LOD of Pb is given as 0.223 ug/dL or 2.23 ug/L.  This means that the average blood level is below the LOD.  Is this correct?  This is a fairly poor analytical technique in that case.  It might better be stated as the % of samples that were below the LOD.

The large Figure 1 and large Figure 2 and the smaller Figure 3 are largely redundant.  Only one is needed because they largely show the same results.  Figure 1, in particular is not needed since all of these analyses are bivariate, whereas the most interesting analysis is multivariate.  Figure 2 is composed of all calculated values, thus with no error bars.  Figure 3 is the smallest and most informative.  My recommendation would be to remove both Figures 1 and 2.

If Figure 1 is included, I have the following concerns.

The caption does a poor job of explaining the figure.  What do the boxes and bars represent?  What are the extra dots in figure B?  These may seem self-evident to the authors but they are not self-evident to the readers, or the reviewers.  Also, why are they so wide?  Instead of 6 per page, there could be 9 per page. 

Line 243-247: This is two sentences combined.  Please make one sentence for curry and one sentence for the vitamins.  You will notice that no OR for curry is given.

If Figure 2 is included, I have the following concerns, in addition to some of those from Figure 1.

There are few zero values on the y-axes.  This creates deceptive impressions of the differences between MetS and nonMetS groups.  This is especially true of the curry graph which absolutely needs to be redrawn to avoid deceiving the reader.  The points and lines on these graphs are all calculated marginal effects.  A reader might mistake them for actual results.  This is most unfortunate.  In the case of CRP, for example, we could be really impressed with the difference between groups until we realize that the per unit adjusted OR is only 1.06.

In Figure 3:

Where is curry?

The units for each of these intakes and serum levels needs to be included so that the reader knows the units for the OR.

It is not clear from the caption with the blue dots represent, the COR or the AOR.

The paragraph that starts on line 243 starts with the line, “After adjustment for comorbidities, the risk of MetS was significantly lower…”  The paragraph that starts on line 262 starts with the line “After adjustment for potential confounders, the adjusted odds ratio followed a similar pattern.”  I am not sure what the difference is between these two descriptions of multiple logistic models are, but they result in different odds ratios.  Would you please explain these differences in more detail so that the reader understands why the ORs in these two paragraphs, and their figures, are different.  The difference between the Cd ORs in the two paragraphs is quite amazing: 113% (OR 2.13; 95% CI, 2.00—2.27, p<0.001), vs 1.01 (0.85, 1.19)

Figure 4 takes up a lot of space to simply show the interaction between age and these variables.  Why is curry here without an age breakdown?  There is very little new information here that could have more easily been presented in a table.  The authors also did not demonstrate statistically the interaction term between MetS risk and each of these independent variables.  As you know, an interaction term must be demonstrated before interactions can be shown.

Author Response

SUNCHON NATIONAL UNIVERSITY

57922 255 Jungang-ro, Suncheon-si,

Jeollanam-do (Seokhyeon-dong), Korea

Tel: 061-750-3132~3 Fax 061-750-3139

E http://www.scnu.ac.kr/ggs/main.do

The Editor

Antioxidant

2021 May 03rd  

RE: Re-submission of a manuscript

Dear Sir/Madam,

On behalf of co-authors, we would like to submit our manuscript entitled, “Effects of antioxidant vitamins, curry consumption and heavy metal levels on the metabolic syndrome with comorbidities: a Korean community-based cross-sectional study” to your journal and request that you consider publishing it as a research article.

In this version, we make a point-by-point response to the comments in the table below. We edited English for the whole manuscript, and we only highlighted in red in our manuscript based on the reviewer’s comments. We hope this version will better than the previous one. Your consideration of our manuscript would be highly appreciated.

Yours sincerely,

Min-Sun Kim, PhD

Department of Pharmacy, College of Pharmacy, Sunchon National University

Sunchon 57922, Republic of Korea

Tel: +821025101635; Email: minsun@scnu.ac.kr

Response to Reviewer 2 Comments

Point 1: Abstract. Lines 13-14: The use of 24-hr recall and food frequency questionnaire are confusing here.  Which is used for what? Line 15: Change “Heavy metals” to Serum heavy metal”.

There is no mention of the cohort used in this study.  Is it cross-sectional; is it adults; is it Korean?

Response 1: Thanks for your comments. We mean that Daily intake of vitamins was measured by 24-h recall, curry consumption was calculated using a food frequency questionnaire. We also added more detailed in the abstract.

Thanks for your suggestion. We changed “Heavy metals” to Serum heavy metal”.

Thanks for your comments. we added more. “A data set of 60,256 Korean aged ≥ 18 years between 2009 and 2017 was used to obtain information on sociodemographic, lifestyle, family history characteristics, MetS, food intake survey, and heavy metals.”

Point 2: Methods

Lines 84-87: Physical activity is poorly defined.  It appears that regular physical activity has 3 requirements: vigorous physical activity, moderate physical activity, and walking. These sum to at least 6 hours per week.  Is this correct?  If this is regular, then what is irregular?  Everybody else with less than 6 hours?

Response 2:

Thanks for your comment. We revised it; we hope it is better than in the previous version. “Physical activity was dichotomized as regular or irregular. Regular physical activity was defined as subjects in (1) vigorous physical activity (running, fast cycling, fast swimming, climbing, football, basketball, singles tennis, squash, rope jumping or occupational or recreational activity involving the carrying of heavy objects), ≥20 minutes per session ≥3 days per week; or subjects in (2) moderate physical activity (slow swimming, volleyball, doubles tennis, or occupational or recreational activity involving the carrying of light objects); ≥30 minutes per session ≥5 days per week; or subjects in (3) walking; ≥30 minutes per session ≥5 days per week.” We also added the reference in this version.

Point 3: Line 147: 25- hydroxy-Vitamin D should be spelled out.

Response 3: Thanks for your comments. we changed Serum 25 (OH)D into “Serum 25 (OH)D”

Point 4: Results

Line 202: What were the age limits, lowest and highest, on KNANES participants?

Response 4: Thanks for your comments. we added more the mean age of participants was 40.8±22.8 (min-max:18-80).

Point 5: Line 207: Underweight is BMI<18.5. Lines 201-216: Since all of these results are shown in Table 1, it is not really necessary to describe them in the text.  You might highlight a few of the more interesting ones, such as curry.  Specifically, you do not need to mention the 5 markers that are part of the MetS.  These are self-evident.

Response 5: Thanks for your comments. We changed BMI<25 into BMI<18.5. We deleted “In addition, levels of total cholesterol, LDL-C, triglyceride, HbA1c, fasting glucose, waist circumference, aspartate aminotransferase (AST), alanine aminotransferase (ALT), systolic and diastolic blood pressure were significantly higher in subjects with MetS compared with subjects without MetS”.

Point 6: Line 228: The value of 125 ug/day for the average intake of vitamin E is suspicious.  This is well below the average of 8.4 mg/day for middle aged Americans and below a value of 20 mg/day for total vitamin E and 3 mg/day for alpha-tocopherol from the following reference for middle aged Koreans.

Nutr Res Pract. 2015 Apr;9(2):192-8. doi: 10.4162/nrp.2015.9.2.192. Epub 2015 Mar 4.“Vitamin E status of 20- to 59-year-old adults living in the Seoul metropolitan area of South Korea Dietary vitamin E and total vitamin E intake (dietary plus supplemental vitamin E) was 17.68 ± 14.34 and 19.55 ± 15.78 mg α-tocopherol equivalents, respectively. The mean daily α-tocopherol, and γ-tocopherol intakes were 3.07 ± 2.27 mg and 5.98 ± 3.74 mg, respectively.”

Response 6: Thank you very much for your comment. We made mistake this is value of retinol rather vitamin E. We changed it this version. “Average daily intakes of total vitamin A and retinol were 692.12±1027.67 µg (95% CI 683.43–700.81) and 125.36 ± 345.14 µg (95% CI 122.44–128.28), respectively”.

Our study used Korea National Health and Nutrition Examination Survey (KNHANES). The antioxidants and total antioxidant capacity (TAC) database consists of 3193 food items consumed by KNHANES subjects from the 2009~2017 survey. The database includes the contents of 43 individual antioxidants and the TAC per food item. The individual antioxidants included four forms of vitamin E (α-tocopherol, β-tocopherol, γ-tocopherol, and δ-tocopherol), with their 31 different components. The following processes were used to measure the intake of vitamin E from dietary supplements. There are two types of dietary supplements: nutraceutical formulations and health functional foods. The components of dietary supplements in nutraceutical formulations were found by searching the Korean Medical Library Engine (www.kmle.co.kr) and Druginfo (www.druginfo.co.kr), both of which report on nutraceutical formulations referenced in the KNHANES. When the products were unavailable on the referenced websites, we investigated their components via the manufacturers' websites or by phone. For health functional foods, we first scanned the websites referenced by the KNHANES; if the referenced websites were not available, we used the Korean Ministry of Food and Drug Safety's health functional food search engine (http://www.foodsafetykorea.go.kr) and manufacturers' websites to find the products and components. Ingredient information for foreign dietary supplements was obtained from the National Institutes of Health's Dietary Supplement Label Database. Finally, if the components could not be identified using these methods, a photograph of the supplement's label or a phone call to the company were used to evaluate the ingredients. However, data is scattered and missing. Therefore, we removed it in this version and added lack of vitamin E in the limitation of our study.

Point 7: Line 229: The average blood value for Pb is given as 2.06 ug/L.  In line 122 of the Methods, the LOD of Pb is given as 0.223 ug/dL or 2.23 ug/L.  This means that the average blood level is below the LOD.  Is this correct?  This is a fairly poor analytical technique in that case.  It might better be stated as the % of samples that were below the LOD.

Response 7: Thanks for your comments. We made mistake about the units of serum Pb ug/dL rather ug/L). Line 229: we changed into “The average serum value for Pb is given as 2.06 ug/dL”. In the method, we added more information about % of samples that were below the LOD. “No sample had a value of below a LOD”.

Point 8: The large Figure 1 and large Figure 2 and the smaller Figure 3 are largely redundant.  Only one is needed because they largely show the same results.  Figure 1, in particular is not needed since all of these analyses are bivariate, whereas the most interesting analysis is multivariate.  Figure 2 is composed of all calculated values, thus with no error bars.  Figure 3 is the smallest and most informative.  My recommendation would be to remove both Figures 1 and 2.

Response 8: Thanks for your comments. We deleted figure 1 and 2.

Point 9: If Figure 1 is included, I have the following concerns.

The caption does a poor job of explaining the figure.  What do the boxes and bars represent?  What are the extra dots in figure B?  These may seem self-evident to the authors but they are not self-evident to the readers, or the reviewers.  Also, why are they so wide?  Instead of 6 per page, there could be 9 per page.

Line 243-247: This is two sentences combined.  Please make one sentence for curry and one sentence for the vitamins.  You will notice that no OR for curry is given.

Response 9: Thanks for your comments. We deleted the figure 1. Thank you very much for your recommendation. We added more information “the risk of MetS was significantly lower in subjects with high curry consumption than subjects with low curry consumption (OR 0.64; 95% CI, 0.58—0.70, p<0.001)”.

Point 10: If Figure 2 is included, I have the following concerns, in addition to some of those from Figure 1.

There are few zero values on the y-axes.  This creates deceptive impressions of the differences between MetS and nonMetS groups.  This is especially true of the curry graph which absolutely needs to be redrawn to avoid deceiving the reader.  The points and lines on these graphs are all calculated marginal effects.  A reader might mistake them for actual results.  This is most unfortunate.  In the case of CRP, for example, we could be really impressed with the difference between groups until we realize that the per unit adjusted OR is only 1.06.

Response 10: Thanks for your comments. We deleted the figure 2.

Point 11: In Figure 3: Where is curry? The units for each of these intakes and serum levels needs to be included so that the reader knows the units for the OR. It is not clear from the caption with the blue dots represent, the COR or the AOR.

Response 11: Thanks for your comments. We added curry consumption and added the caption with the blue dots represent for AOR.

Point 12: The paragraph that starts on line 243 starts with the line, “After adjustment for comorbidities, the risk of MetS was significantly lower…”  The paragraph that starts on line 262 starts with the line “After adjustment for potential confounders, the adjusted odds ratio followed a similar pattern.”  I am not sure what the difference is between these two descriptions of multiple logistic models are, but they result in different odds ratios.  Would you please explain these differences in more detail so that the reader understands why the ORs in these two paragraphs, and their figures, are different. The difference between the Cd ORs in the two paragraphs is quite amazing: 113% (OR 2.13; 95% CI, 2.00—2.27, p<0.001), vs 1.01 (0.85, 1.19)

Response 12: Thanks for your comments. line 243, we main that we only adjustment for comorbidities. Line 262, we main that we adjusted for monthly household income, residential areas, energy intake, occupation, sex, family history of CVDs, family history of diabetes mellites, family history of hyperlipidemia, BMI group, smoking status, cotinine verified smokers, high-risk drinking, physical activity, education level, hypertension, dyslipidemia, type 2 diabetes, stroke, myocardial infarction or angina, myocardial infarction, angina, arthritis, osteoarthritis, rheumatoid arthritis, kidney failure, depression, thyroid disease, and asthma, green-vegetable, white-vegetable, fruit.

We changed into“After adjustment for potential confounders including monthly household income, residential areas, energy intake, occupation, sex, family history of CVDs, family history of diabetes mellites, family history of hyperlipidemia, BMI group, smoking status, cotinine verified smokers, high-risk drinking, physical activity, education level, hypertension, dyslipidemia, type 2 diabetes, stroke, myocardial infarction or angina, myocardial infarction, angina, arthritis, osteoarthritis, rheumatoid arthritis, kidney failure, depression, thyroid disease, and asthma, green-vegetable, white-vegetable, fruit, the adjusted odds ratio followed a similar pattern.”

Point 13: Figure 4 takes up a lot of space to simply show the interaction between age and these variables.  Why is curry here without an age breakdown?  There is very little new information here that could have more easily been presented in a table.  The authors also did not demonstrate statistically the interaction term between MetS risk and each of these independent variables.  As you know, an interaction term must be demonstrated before interactions can be shown.

Response 13: Figure 4 takes up a lot of space to simply show the interaction between age and these variables.  Why is curry here without an age breakdown?  There is very little new information here that could have more easily been presented in a table.  The authors also did not demonstrate statistically the interaction term between MetS risk and each of these independent variables.  As you know, an interaction term must be demonstrated before interactions can be shown.          

Response 13: Thanks for your comments. In this study, we did not evaluate the interaction between MetS and each of these independent variables in Figure 4. We aimed to visualize the moderating effect of the MetS, marginal effects were performed using the results of logistic regression analysis after adjusting for potential variables.

Curry consumption is dichotomous variable. Therefore, we only draw the figure with probability of MetS without age breakdown.

Here is code for curry consumption:

Logistic METS i.curry_consum1 i.household_income i.area energy i.age_groupmet i.occupation i.sex_status i.FH_CVD i.FH_DM i.FH_hyperlipid i.BMI_group i.current_smoker i.HR_drinking i.physical_act i.edu_level i.hypertension i.dyslipidemia i.Diabetes i.stroke i.myo_angina i.myocardial i.angina i.Arthritis i.Osteoarthritis i.Rheumatoid_arthritis i.kidney_failure i.depression i.Thyroid i.Asthma

margins

margins curry_consum1

marginsplot, recast(bar)

Here is code for margin effect of serum Pb:

logistic METS HE_Pb i.household_income i.area energy i.age_groupmet i.occupation i.sex_status i.FH_CVD i.FH_DM i.FH_hyperlipid i.BMI_group i.current_smoker i.HR_drinking i.physical_act i.edu_level i.hypertension i.dyslipidemia i.Diabetes i.stroke i.myo_angina i.myocardial i.angina i.Arthritis i.Osteoarthritis i.Rheumatoid_arthritis i.kidney_failure i.depression i.Thyroid i.Asthma i.greenvegetable i.whitevegetable i.fruit

qui margins age_groupmet , at( HE_Pb =(0(1)5)) atmean

marginsplot, noci plot1opt(lp(dash))

Reviewer 3 Report

Comments to authors:

First, thank you for opportunity to review the manuscript entitled “Effects of antioxidant vitamins, curry consumption and heavy metal levels on the metabolic syndrome with comorbidities: a Korean community-based cross-sectional study”. This research includes cross-sectional analyses to boarding factors that could be related to metabolic syndrome. However, I observe critical concerns regard this study.

On the one hand, the items considered seem to be randomly selected. There is not a logical relationship between them. In addition, throughout the text, authors describe differently the factors studied. For example, authors talk about curry (from rice dishes), vitamins (sometimes it is not clear which vitamins and if these are evaluated from intake or from biological samples) and heavy metals (Pb, Hg and Cd). However, they also report results about clinical measures (such as CRP) and study the item “vegetable intake” (I am not agreeing with the classification of vegetables). The study has not an organized structure neither logical sense. For example, why curry? Is it this measured reliable? Are there any criteria to select of these specifical items? I don`t get to understand the sense of this study. In addition, methods to quantified intake of vitamins, vegetables and curry are different (different FFQ, 24-h recalls…), it is should be homogenized.

On the other hand, major reviews of English should be performed on the manuscript. There are critical errors that difficult the lecture.

Author Response

SUNCHON NATIONAL UNIVERSITY

57922 255 Jungang-ro, Suncheon-si,

Jeollanam-do (Seokhyeon-dong), Korea

Tel: 061-750-3132~3 Fax 061-750-3139

E http://www.scnu.ac.kr/ggs/main.do

The Editor

Antioxidant

2021 May 03rd  

RE: Re-submission of a manuscript

Dear Sir/Madam,

On behalf of co-authors, we would like to submit our manuscript entitled, “Effects of antioxidant vitamins, curry consumption and heavy metal levels on the metabolic syndrome with comorbidities: a Korean community-based cross-sectional study” to your journal and request that you consider publishing it as a research article.

In this version, we make a point-by-point response to the comments in the table below. We edited English for the whole manuscript, and we only highlighted in red in our manuscript based on the reviewer’s comments. We hope this version will better than the previous one. Your consideration of our manuscript would be highly appreciated.

Yours sincerely,

Min-Sun Kim, PhD

Department of Pharmacy, College of Pharmacy, Sunchon National University

Sunchon 57922, Republic of Korea

Tel: +821025101635; Email: minsun@scnu.ac.kr

Response to Reviewer 3 Comments

Point 1: First, thank you for opportunity to review the manuscript entitled “Effects of antioxidant vitamins, curry consumption and heavy metal levels on the metabolic syndrome with comorbidities: a Korean community-based cross-sectional study”. This research includes cross-sectional analyses to boarding factors that could be related to metabolic syndrome. However, I observe critical concerns regard this study.

 On the one hand, the items considered seem to be randomly selected. There is not a logical relationship between them. In addition, throughout the text, authors describe differently the factors studied. For example, authors talk about curry (from rice dishes), vitamins (sometimes it is not clear which vitamins and if these are evaluated from intake or from biological samples) and heavy metals (Pb, Hg and Cd). However, they also report results about clinical measures (such as CRP) and study the item “vegetable intake” (I am not agreeing with the classification of vegetables). The study has not an organized structure neither logical sense. For example, why curry? Is it this measured reliable? Are there any criteria to select of these specifical items? I don`t get to understand the sense of this study. In addition, methods to quantified intake of vitamins, vegetables and curry are different (different FFQ, 24-h recalls…), it is should be homogenized.

Response 1: Thanks for your great comments. We would like to summary our purpose of this study. As known, the prevalence and incidence of MetS are increasing in Korea. In addition to lifestyle and genetic factors, the effects of heavy metal on the risk factors of MetS. Furthermore, increasing evidence shows that vitamin supplementation could reverse CVDs, and diabetes, and mental illness. Based on this evidence, we aimed to assess the association between MetS and risk factors including heavy metals, biomarkers (hs-CRP) and food intake including daily vitamin take from food, curry consumption (curry rice). First, we described the characteristics of MetS, food intake, and heavy metal. Second, we assessed the association between MetS, heavy metals, biomarkers (hs-CRP), and food intake. Subsequently, we visualized the effects of heavy metals and food intake by margin effects. In this version, we revised it based on the order we described and two other reviewers, we hope that this version will better than the previous one.

In this study, vitamin intake is totally from food intake (not from biological samples). Vitamin intake was collected from 24h recalls.

Green or white vegetable, fruit as well as curry were collected from FFQ. We revised it in the method. In terms of vegetable intake, we revised it into “green vegetable, white vegetable, fruit.

In this study, the vegetable intake was categorized based on the previous study. In this version, we also cited the reference in our study. Reference: Nutrition. 2015 Jan;31(1):111-8. doi: 10.1016/j.nut.2014.05.011. Epub 2014 Jun 18.

The curry consumption was estimated using the KNHANES FFQ data. Curry rice was the only curry-related food among the food items surveyed. The curry consumption was used based on the previous study (Reference: Nutr Res Pract. 2016 Apr;10(2):212-20. doi: 10.4162/nrp.2016.10.2.212. Epub 2016 Jan 28.). In the version, we also cited in our paper.

We also summarized our purpose of this study by using the marginal effects to visualize the moderating effect of the MetS.

Point 2: On the other hand, major reviews of English should be performed on the manuscript. There are critical errors that difficult the lecture.

Response 2: Thanks for your valuable comments. First, we tried to check grammatical structure. After that, we sent this manuscript for our colleague to proofread. We hope that this version is better than the previous one.

Round 2

Reviewer 1 Report

I have no comments.

Author Response

Dear Sir/Madam,

We hope that this finds you well and that you and your family have been keeping you safe over these past months.   Thank you very much for giving us great comments and a chance to publish our paper.  

Yours sincerely,
Min-Sun Kim, PhD
Corresponding Author

Department of Pharmacy, College of Pharmacy, Sunchon National University
Sunchon 57922, Republic of Korea
Tel: +821025101635; Email: minsun@scnu.ac.kr

Reviewer 2 Report

The authors have worked hard to improve the English language and style on this paper and have satisfied the concerns that I had in the first review.  I have no more concerns.

Author Response

(The authors gave the same response as above.)

Reviewer 3 Report

The manuscript entitled “Effects of antioxidant vitamins, curry consumption and heavy metal levels on the metabolic syndrome with comorbidities: a Korean community-based cross-sectional study” have been improved, mainly the English grammatical. However, I continue having some concerns about this work:

  1. Authors say “biomarkers” throughout manuscript, but they don`t define these biomarkers. I think that they could refer to “hs-CRP”. However, it is not clear or correct to use “biomarker” in this case. “hs-CRP” is a pro-inflammatory protein, but this is a marker of any inflammation process.
  2. Vitamin intake should be defined before in the manuscript, but only some of them are evaluated. In addition, serum liposoluble vitamins are described, but it does not appear in the abstract. The information should be homogenized throughout manuscript.
  3. Line 35: “fatty, high-saturated” should be replaced by “high-saturated fat”
  4. Line 35: I think that author pretend to say “high-energy diets” instead of “low-energy diets”
  5. Lines 23, 49, 321 and 356: authors refer to supplementation, but they study dietary intake from food. This term seems confuse.
  6. Line 50: authors talk about curcumin, but they only evaluated the curry intake (from curry rice). Why they not considered curcumin? Why is the reason to select this spice? It is curious that consume habitual only is a 1% and they selected particularly this condiment. In addition, quantification from curry rice could be not robust.
  7. Lines 52-54: It seems a result instead an objective. I suggest replaced it by: In this study, we study whether an increased intake of vitamins and curry reduce the risk of MetS in the Korean population with or without various NCDs, and whether the serum heavy metals and hs-CRP are positively associated with the risk of MetS.
  8. Line 52: curry is not consumed daily.
  9. Line 74: “year?” should be completed with a date.
  10. Lines 102, 105: Height, weight, waist circumference, blood pressure.
  11. In the section 2.3. it is not explained the assay of HbA1c.
  12. Line 121: “No sample had a value below LOD” should be in the section of results.
  13. Line 136, 138: values to women, but not to men, are reported. Both sexes should be considered due to population included women and men.
  14. Lines 141-144: Description of methodology regarding hs-CRP should be in another subsection different to 2.6. Metabolic Syndrome. Hs-CRP is related to inflammation status, but it is not a criterion to diagnose MetS.
  15. Line 147, 151: An explanation of methodology or at less references about the method used should be reported.
  16. Regarding vegetable classification, I think that “white vegetables” should be called “other vegetables” because this classification this is not clear. Usually, authors use different classification to vegetable group, but in my opinion vegetables as tomato, carrot and pumpkin are not “white vegetable”, they are red/orange vegetables (or carotene-vegetables).
  17. Line 224: p-value should be reported.
  18. Line 225: It is corrected that vitamin A and E where higher in subjects with MetS?
  19. Lines 342-343: It is not clear. I suggest to authors to define clearer what vitamins are evaluated in serum/intake.
  20. I think that authors need to review the methodology. For example, they don`t describe the method used to measure the physical activity. They don`t report the moment of the collection of 24-recalls and food frequency questionnaire.

Round 3

Reviewer 3 Report

The last version of the manuscript has been improved.